# Calibrating Deep Ensemble through Functional Variational Inference

**Zhijie Deng**                                                                          *zhijied@sjtu.edu.cn*
*Qing Yuan Research Institute, SEIEE, Shanghai Jiao Tong University*

**Feng Zhou**                                                                           *feng.zhou@ruc.edu.cn*
*Center for Applied Statistics and School of Statistics, Renmin University of China*

**Jianfei Chen**                                                                        *jianfeic@tsinghua.edu.cn*
*Dept. of Comp. Sci. & Tech., Tsinghua University*

**Guoqiang Wu**                                                                         *guoqiangwu90@gmail.com*
*School of Software, Shandong University*

**Jun Zhu**                                                                             *dcszj@tsinghua.edu.cn*
*Dept. of Comp. Sci. & Tech., Tsinghua University*

**Reviewed on OpenReview:** *https://openreview.net/forum?id=uvPnTWMLll*

## Abstract

Deep Ensemble (DE) is an effective and practical uncertainty quantification approach in deep learning. The uncertainty of DE is usually manifested by the functional inconsistency among the ensemble members, which, yet, originates from unmanageable randomness in the initialization and optimization of neural networks (NNs), and may easily collapse in specific cases. To tackle this issue, we advocate characterizing the functional inconsistency with the empirical covariance of the functions dictated by the ensemble members, and defining a Gaussian process (GP) with it. We perform functional variational inference to tune such a probabilistic model w.r.t. training data and specific prior beliefs. This way, we can explicitly manage the uncertainty of the ensemble of NNs. We further provide strategies to make the training efficient. The proposed approach achieves better uncertainty quantification than DE and its variants across diverse scenarios, while consuming only marginally added training cost compared to standard DE. The code is available at `https://github.com/thudzj/DE-GP`.

## 1 Introduction

Bayesian treatment of deep neural networks (DNNs) is promised to enjoy principled Bayesian uncertainty while unleashing the capacity of DNNs, with Bayesian neural networks (BNNs) as popular examples (MacKay, 1992; Hinton & Van Camp, 1993; Neal, 1995; Graves, 2011). Nevertheless, despite the surge of advance in BNNs (Louizos & Welling, 2016; Zhang et al., 2018), many existing BNNs still face obstacles in accurate and scalable inference (Sun et al., 2019), and exhibit limitations in uncertainty quantification and out-of-distribution robustness (Ovadia et al., 2019).

Deep Ensemble (DE) (Lakshminarayanan et al., 2017) is an effective and practical method for uncertainty quantification, which assembles multiple independently trained DNNs for prediction. DE presents higher flexibility and effectiveness than typical BNN methods. Practitioners tend to interpret the functional inconsistency among the ensemble members, say, the disagreement among their predictions, as a proxy of DE's uncertainty (Smith & Gal, 2018). However, the functional inconsistency stems from the unmanageable randomness in DNN initialization and stochastic gradient descent (SGD), thus is likely to collapse in specific cases (see Fig. 1). To fix this issue, recent works like randomised MAP sampling (RMS) (Lu & Van Roy,

2017; Osband et al., 2018; Pearce et al., 2020; Ciosek et al., 2019) and NTKGP (He et al., 2020) refine DE in the spirit of "sample-then-optimize" (Matthews et al., 2017), but they often rely on restrictive assumptions like Gaussian likelihoods, linearized/infinite-width models, etc., having difficulties to generalize.

This paper aims to calibrate the uncertainty of DE in a more feasible way. We first reveal that there exists a gap between the training and test of DE: the functional inconsistency in DE has not been properly adapted w.r.t. training data and prior uncertainty but is used to quantify post data uncertainty in the test phase. To bridge the gap, we propose to incorporate functional inconsistency into modeling and training explicitly. Viewing the ensemble members as a set of basis functions, the functional inconsistency can be formally described by their empirical covariance, which, along with their mean, specify a Gaussian process (GP). We perform functional variational inference (fVI) (Sun et al., 2019) to holistically tune such a GP (dubbed as DE-GP), making it approximate the function-space Bayesian posterior, during which the uncertainty of the NN ensemble is explicitly calibrated.

In essence, DE-GP adds to the line of works that build low-rank approximations to the Bayesian posterior in function space (Deng et al., 2022) or weight space (Maddox et al., 2019; Izmailov et al., 2020; Dusenberry et al., 2020). The success of these works signifies that in the case of DNNs, low-rank approximations to Bayesian posteriors can reasonably conjoin performance and efficiency. However, distinct from them, we confine the model family to expressive DNN ensembles, which significantly boosts the flexibility of the approximate posterior. Compared to related works on calibrating DE's uncertainty (e.g., He et al., 2020), our approach can handle classification problems directly, without casting them into regression ones, due to the use of fVI.

Technically, we adopt a prior also in the GP family, and then the gradients of the KL divergence between DE-GP and the GP prior involved in fVI can be easily estimated without relying on complicated gradient estimators (Shi et al., 2018b). We provide recipes to make the training even faster, thus the additional computation overhead of DE-GP upon DE is minimal.

We study the behavior of DE-GP on diverse benchmarks. Empirically, DE-GP outperforms DE and its variants on various regression datasets and presents superior uncertainty estimates and out-of-distribution robustness without compromising accuracy in standard image classification tasks. DE-GP also shows promise in solving contextual bandit problems, where uncertainty plays a vital role in guiding exploration.

## 2 Related Work

Bayesian treatment of DNNs is an emerging topic yet with a long history (Mackay, 1992; Hinton & Van Camp, 1993; Neal, 1995; Graves, 2011). BNNs can be learned by variational inference (Blundell et al., 2015; Louizos & Welling, 2016; Zhang et al., 2018; Khan et al., 2018; Deng et al., 2020), Laplace approximation (Mackay, 1992; Ritter et al., 2018), Markov chain Monte Carlo (Welling & Teh, 2011; Chen et al., 2014), Monte Carlo dropout (Gal & Ghahramani, 2016), etc. To avoid the difficulties of posterior inference in weight space, some works advocate performing functional Bayesian inference (Sun et al., 2019; Rudner et al., 2021; Wang et al., 2019). Rudner et al. (2022) then improve such approaches by developing a simple finite-sample estimator of the involved function-space KL divergence. In function space, BNNs of infinite or even finite width equal to GPs (Neal, 1996; Lee et al., 2018; Novak et al., 2018; Khan et al., 2019), which provides supports for constructing an approximate posterior in the form of GP.

DE (Lakshminarayanan et al., 2017) is a practical approach to uncertainty quantification and has shown promise in diverse scenarios (Ovadia et al., 2019). On one hand, DE has been interpreted as a method that approximates the Bayesian posterior predictive (Wilson & Izmailov, 2020); the approximation quality is empirically studied by Izmailov et al. (2021). On the other hand, some works argue that DE lacks a principled Bayesian justification (Pearce et al., 2020; He et al., 2020) and propose to refine DE by specific principles. For example, RMS (Lu & Van Roy, 2017; Osband et al., 2018; Pearce et al., 2020; Ciosek et al., 2019) regularizes the ensemble members towards randomised priors to obtain posterior samples, but it typically assumes linear data likelihood which is impractical for deep models and classification tasks. He et al. (He et al., 2020) propose to add a randomised function to each ensemble member to realize a function-space Bayesian interpretation, but the method is asymptotically exact in the infinite width limit and is limited to regressions. By contrast, DE-GP calibrates DE's uncertainty without restrictive assumptions. A concurrent work proposes to add

a repulsive term to DE (D'Angelo & Fortuin, 2021) based on the principle of particle-optimization based variational inference (POVI) (Liu & Wang, 2016), but the method relies on less scalable gradient estimators for optimization.

## 3 Motivation

Assume a dataset $\mathcal{D} = (\mathbf{X}, \mathbf{Y}) = \{(\boldsymbol{x}_i, \boldsymbol{y}_i)\}_{i=1}^n$, with $\boldsymbol{x}_i \in \mathcal{X}$ as data and $\boldsymbol{y}_i$ as $C$-dimensional targets. We can deploy a DNN $g(\cdot, \boldsymbol{w}) : \mathcal{X} \to \mathbb{R}^C$ with weights $\boldsymbol{w}$ for fitting. Despite impressive performance, the regularly trained DNNs are prone to over-confidence, making it hard to decide how certain they are about the predictions. Lacking the ability to reliably quantify predictive uncertainty is unacceptable for realistic decision-making scenarios.

A principled mechanism for uncertainty quantification in deep learning is to incorporate Bayesian treatment to reason about Bayesian uncertainty. The resulting models are known as BNNs. In BNNs, $\boldsymbol{w}$ is treated as a random variable. Given some prior beliefs $p(\boldsymbol{w})$, we chase the posterior $p(\boldsymbol{w}|\mathcal{D})$. In practice, it is intractable to analytically compute the true posterior $p(\boldsymbol{w}|\mathcal{D})$ due to the high non-linearity of DNNs, so some approximate posterior $q(\boldsymbol{w})$ is usually found by techniques like variational inference (Blundell et al., 2015), Laplace approximation (Mackay, 1992), Monte Carlo (MC) dropout (Gal & Ghahramani, 2016), etc.

BNNs perform marginalization to predict for new data $\boldsymbol{x}^*$ (a.k.a. posterior predictive):

$$p(y|\boldsymbol{x}^*, \mathcal{D}) = \mathbb{E}_{p(\boldsymbol{w}|\mathcal{D})} p(y|\boldsymbol{x}^*, \boldsymbol{w}) \approx \mathbb{E}_{q(\boldsymbol{w})} p(y|\boldsymbol{x}^*, \boldsymbol{w}) \approx \frac{1}{S} \sum_{s=1}^S p(y|\boldsymbol{x}^*, \boldsymbol{w}_s), \tag{1}$$

where $\boldsymbol{w}_s \sim q(\boldsymbol{w}), s = 1, ..., S$. Nonetheless, most of the existing BNN approaches face obstacles in precise posterior inference due to non-trivial and convoluted posterior dependencies (Louizos & Welling, 2016; Zhang et al., 2018; Shi et al., 2018a; Sun et al., 2019), and deliver unsatisfactory uncertainty quantification performance (Ovadia et al., 2019).

As a practical uncertainty quantification method, Deep Ensemble (DE) (Lakshminarayanan et al., 2017) deploys a set of $M$ DNNs $\{g(\cdot, \boldsymbol{w}_i)\}_{i=1}^M$ to interpret the data from different angles. The ensemble members are independently trained under deterministic learning principles like maximum likelihood estimation (MLE) and maximum a posteriori (MAP) (we refer to the resulting models as DE and regularized DE (rDE) respectively):

$$\max_{\boldsymbol{w}_1, ..., \boldsymbol{w}_M} \frac{1}{M} \sum_{i=1}^M \log p(\mathcal{D}|\boldsymbol{w}_i),$$

$$\max_{\boldsymbol{w}_1, ..., \boldsymbol{w}_M} \frac{1}{M} \sum_{i=1}^M [\log p(\mathcal{D}|\boldsymbol{w}_i) + \log p(\boldsymbol{w}_i)]. \tag{2}$$

The randomness in model initialization and SGD diversifies the ensemble members, driving them to explore distinct modes of the non-convex loss landscape of DNNs (Fort et al., 2019; Wilson & Izmailov, 2020). This not only boosts the ensemble performance but also renders DE an approach to uncertainty quantification (Lakshminarayanan et al., 2017; Ovadia et al., 2019). The functional inconsistency among the ensemble members is usually interpreted as a proxy of DE's uncertainty (Smith & Gal, 2018). Yet, we question its effectiveness given that the functional inconsistency is caused by the aforementioned uncontrollable randomness instead of explicit Bayesian inference. To check if this is the case, we evaluate DE and rDE on a 1-D regression problem. We choose neural network Gaussian process (NN-GP) (Neal, 1996) posteriors as a gold standard as they equal to infinite-width well-trained BNNs and can be analytically estimated. We depict the results in Fig. 1.

The results echo our concerns on DE's uncertainty. It is evident that (*i*) DE and rDE collapse to a single model in the extreme linear case (i.e., without hidden layers), because the loss surface is convex w.r.t. the model parameters; (*ii*) DE and rDE reveal minimal uncertainty in in-distribution regions, although there is severe data noise; (*iii*) the uncertainty of DE and rDE deteriorates as the model size increases regardless of whether there is over-fitting.

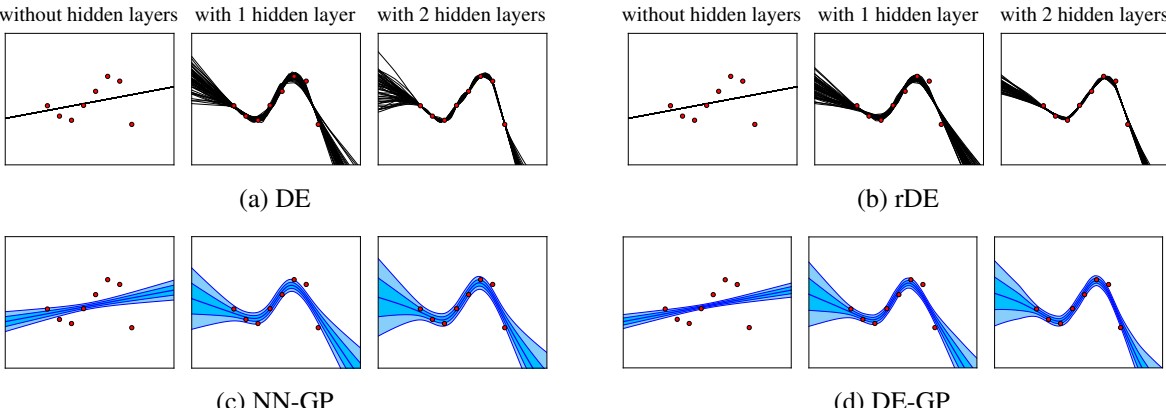

Figure 1:   1-D regression on $y = \sin 2x + \epsilon, \epsilon \sim \mathcal{N}(0, 0.2)$. We use 50 multilayer perceptrons (MLPs) for ensemble and experiment on 3 architectures (with different numbers of hidden layers). The weight-space priors for rDE, DE-GP, and NN-GP are Gaussian distributions. Red dots refer to the training data. Black lines for DE and rDE refer to the predictions of the ensemble members. Dark blue curves and shaded regions for DE-GP and NN-GP refer to mean predictions and uncertainty. Compared to NN-GP, DE and rDE suffer from over-confidence and less calibrated uncertainty estimates. DE-GP can address these issues.

The functional inconsistency in DE has not been properly adapted w.r.t. training data and prior uncertainty, but is used to quantify post data uncertainty. Such a gap may be the cause of the unreliability issue of DE's uncertainty. Having identified this, we propose to incorporate the functional inconsistency into modeling, and perform functional Bayesian inference to tune the whole model. We describe how to realize this below.

## 4   Methodology

This section provides the details for the modeling, inference, and training of DE-GP. We impress the readers in advance with the results of DE-GP on the aforementioned regression problem as shown in Fig. 1.

### 4.1   Modeling

Viewing the ensemble members $\{g(\cdot, \boldsymbol{w}_i)\}_{i=1}^M$ as a set of basis functions, the functional inconsistency among them can be formally represented by the empirical covariance:

$$k(\boldsymbol{x}, \boldsymbol{x}') := \frac{1}{M} \sum_{i=1}^M \left(g_i(\boldsymbol{x}) - m(\boldsymbol{x})\right) \left(g_i(\boldsymbol{x}') - m(\boldsymbol{x}')\right)^\top, \tag{3}$$

where $g_i$ refers to $g(\cdot, \boldsymbol{w}_i)$ and $m(\boldsymbol{x}) := \frac{1}{M} \sum_{i=1}^M g_i(\boldsymbol{x})$. From the definition, $k$ is a matrix-valued kernel, with values in the space of $C \times C$ matrices.

Then, the incorporation of functional inconsistency amounts to building a model with $k(\boldsymbol{x}, \boldsymbol{x}')$. Naturally, the DE-GP comes into the picture, defined as $\mathcal{GP}(f|m(\boldsymbol{x}), k(\boldsymbol{x}, \boldsymbol{x}'))$. Given that $k(\boldsymbol{x}, \boldsymbol{x}')$ is of low rank, we opt to add a small scaled identity matrix $\lambda \mathbf{I}_C$[1] upon $k(\boldsymbol{x}, \boldsymbol{x}')$ to avoid singularity. Unless specified otherwise, we refer to the resulting covariance kernel as $k(\boldsymbol{x}, \boldsymbol{x}')$ in the following.

The variations in $k(\boldsymbol{x}, \boldsymbol{x}')$ are confined to having up to $M - 1$ rank, echoing the recent investigations showing that low-rank approximate posteriors for deep models conjoin effectiveness and efficiency (Deng et al., 2022; Maddox et al., 2019; Izmailov et al., 2020; Dusenberry et al., 2020). Akin to typical deep kernels (Wilson et al., 2016), the DE-GP kernels are highly flexible, and may automatically discover the underlying structures of high-dimensional data without manual participation.

---

[1] $\mathbf{I}_C$ refers to the identity matrix of size $C \times C$.

## 4.2 Inference

DE-GP is a Bayesian model instead of a point estimate, so we tune it by approximate Bayesian inference principle – take it as a parametric approximate posterior, i.e.,

$$q(f) = \mathcal{GP}(f|m(\boldsymbol{x}), k(\boldsymbol{x}, \boldsymbol{x}')), \tag{4}$$

and leverage fVI (Sun et al., 2019) to push it towards the true posterior over functions associated with specific priors.

**Prior** In fVI, we can freely choose a distribution over functions (i.e., a stochastic process) as the prior. Nonetheless, to avoid unnecessary convoluted gradient estimation, we adopt a prior also in the GP family:

$$p(f) = \mathcal{GP}(f|0, k_p(\boldsymbol{x}, \boldsymbol{x}')), \tag{5}$$

where the prior mean is assumed to be zero by convention.

On the one hand, this way, we can easily incorporate prior knowledge by specifying a prior kernel which encodes appropriate structures like data similarity or periodicity for the sake of good data fitting. On the other hand, we can flexibly combine basic kernels by simple multiplication or addition to increase the expressiveness of the prior.

Typically, most well-evaluated kernels, dubbed as $\hat{k}_p(\boldsymbol{x}, \boldsymbol{x}')$, are scalar-valued, but we can extend them to be matrix-valued by enforcing isotropy via $k_p(\boldsymbol{x}, \boldsymbol{x}') = \hat{k}_p(\boldsymbol{x}, \boldsymbol{x}')\mathbf{I}_C$.

**fELBO** We maximize the functional Evidence Lower BOund (fELBO) (Sun et al., 2019) to perform fVI:

$$\max_{q(f)} \mathbb{E}_{q(f)}[\log p(\mathcal{D}|f)] - D_{\mathrm{KL}}[q(f)\|p(f)]. \tag{6}$$

Notably, there is a KL divergence between two GPs, which, on its own, is challenging to cope with. Fortunately, as proved by Sun et al. (2019), we can take the KL divergence between the marginal distributions of function evaluations as a substitute for it, giving rise to a more tractable objective:

$$\mathcal{L} = \mathbb{E}_{q(f)}\Big[ \sum_{(\boldsymbol{x}_i, \boldsymbol{y}_i) \in \mathcal{D}} \log p(\boldsymbol{y}_i|f(\boldsymbol{x}_i)) \Big] - D_{\mathrm{KL}}\Big[q(\mathbf{f}^{\tilde{\mathbf{X}}})\|p(\mathbf{f}^{\tilde{\mathbf{X}}})\Big], \tag{7}$$

where $\tilde{\mathbf{X}}$ denotes a measurement set including all training inputs $\mathbf{X}$, and $\mathbf{f}^{\tilde{\mathbf{X}}}$ is the concatenation of the vectorized outputs of $f$ for $\tilde{\mathbf{X}}$, i.e., $\mathbf{f}^{\tilde{\mathbf{X}}} \in \mathbb{R}^{|\tilde{\mathbf{X}}|C}$.[2]

It has been recently shown that fELBO is often ill-defined and may lead to several pathologies as the KL divergence in function space is infinite (Burt et al., 2020). Yet, the results in Fig. 1, which serve as a posterior approximation quality check, reflect that fELBO is empirically effective for tuning DE-GP.

## 4.3 Training

We outline the training procedure in Algorithm 1, and elaborate some details below.

**Mini-batch training** DE-GP should proceed by mini-batch training when facing large data. At each step, we manufacture a stochastic measurement set with a mini-batch $\mathcal{D}_s = (\mathbf{X}_s, \mathbf{Y}_s)$ from the training data $\mathcal{D}$ and some random samples $\mathbf{X}_\nu$ from a continuous distribution (e.g., a uniform distribution) $\nu$ supported on $\mathcal{X}$. Then, we adapt the objective defined in Eq. (7) to:

$$\max_{\boldsymbol{w}_1, \ldots, \boldsymbol{w}_M} \mathcal{L} = \max_{\boldsymbol{w}_1, \ldots, \boldsymbol{w}_M} \mathbb{E}_{f \sim q(f)}\Big[ \sum_{(\boldsymbol{x}_i, \boldsymbol{y}_i) \in \mathcal{D}_s} \log p(\boldsymbol{y}_i|f(\boldsymbol{x}_i)) \Big] - \alpha D_{\mathrm{KL}}\Big[q(\mathbf{f}^{\tilde{\mathbf{X}}_s})\|p(\mathbf{f}^{\tilde{\mathbf{X}}_s})\Big], \tag{8}$$

where $\tilde{\mathbf{X}}_s$ indicates the union of $\mathbf{X}_s$ and $\mathbf{X}_\nu$. Instead of fixing $\alpha$ as 1, we opt to fix the hyper-parameters specifying the prior $p(f)$, but to tune the positive coefficient $\alpha$ to better trade off between data evidence and

---

[2]We use $|\tilde{\mathbf{X}}|$ to notate the size of a set $\tilde{\mathbf{X}}$.

---

**Algorithm 1** The training of DE-GP

---

1: **Input:** $\mathcal{D}$: dataset; $\{g(\cdot, \boldsymbol{w}_i)\}_{i=1}^M$: a deep ensemble; $k_p$: prior kernel; $\nu$: distribution for sampling extra measurement points; $U$: number of MC samples for estimating expected log-likelihood
2: **while** not converged **do**
3:      $\mathcal{D}_s = (\mathbf{X}_s, \mathbf{Y}_s) \subset \mathcal{D}$, $\mathbf{X}_\nu \sim \nu$, $\tilde{\mathbf{X}}_s = \{\mathbf{X}_s, \mathbf{X}_\nu\}$
4:      $\mathbf{g}_i^{\tilde{\mathbf{X}}_s} = g(\tilde{\mathbf{X}}_s, \boldsymbol{w}_i)$, $i = 1, ..., M$
5:      $\mathbf{m}^{\tilde{\mathbf{X}}_s} = \frac{1}{M} \sum_i \mathbf{g}_i^{\tilde{\mathbf{X}}_s}$
6:      $\mathbf{k}^{\tilde{\mathbf{X}}_s, \tilde{\mathbf{X}}_s} = \frac{1}{M} \sum_{i=1}^M (\mathbf{g}_i^{\tilde{\mathbf{X}}_s} - \mathbf{m}^{\tilde{\mathbf{X}}_s})(\mathbf{g}_i^{\tilde{\mathbf{X}}_s} - \mathbf{m}^{\tilde{\mathbf{X}}_s})^\top + \lambda \mathbf{I}_{|\tilde{\mathbf{X}}_s|C}$
7:      $\mathbf{k}_p^{\tilde{\mathbf{X}}_s, \tilde{\mathbf{X}}_s} = k_p(\tilde{\mathbf{X}}_s, \tilde{\mathbf{X}}_s)$
8:      $\mathcal{L}_1 = \frac{1}{U} \sum_{i=1}^U \sum_{(\boldsymbol{x}, \boldsymbol{y}) \in \mathcal{D}_s} \log p(\boldsymbol{y}|\mathbf{f}_i(\boldsymbol{x}))$, $\mathbf{f}_i \sim \mathcal{N}(\mathbf{m}^{\tilde{\mathbf{X}}_s}, \mathbf{k}^{\tilde{\mathbf{X}}_s, \tilde{\mathbf{X}}_s})$
9:      $\mathcal{L}_2 = D_{\text{KL}}[\mathcal{N}(\mathbf{m}^{\tilde{\mathbf{X}}_s}, \mathbf{k}^{\tilde{\mathbf{X}}_s, \tilde{\mathbf{X}}_s}) \| \mathcal{N}(\mathbf{0}, \mathbf{k}_p^{\tilde{\mathbf{X}}_s, \tilde{\mathbf{X}}_s})]$
10:      $\boldsymbol{w}_i = \boldsymbol{w}_i + \eta \nabla_{\boldsymbol{w}_i}(\mathcal{L}_1 - \alpha \mathcal{L}_2)$, $i = 1, ..., M$
11: **end while**

---

prior regularization. When tuning $\alpha$, we intentionally set it as large as possible to avoid colder posteriors and worse uncertainty estimates.

The importance of the incorporation of extra measurement points $\mathbf{X}_\nu$ depends on the data and the problem at hand. We perform a study on the aforementioned 1-D regression without incorporating $\mathbf{X}_\nu$ in Appendix, and the results are still seemingly promising.

The marginal distributions in the KL are both multivariate Gaussians, i.e.,

$$q(\mathbf{f}^{\tilde{\mathbf{X}}_s}) = \mathcal{N}(\mathbf{f}^{\tilde{\mathbf{X}}_s} | \mathbf{m}^{\tilde{\mathbf{X}}_s}, \mathbf{k}^{\tilde{\mathbf{X}}_s, \tilde{\mathbf{X}}_s}), \quad p(\mathbf{f}^{\tilde{\mathbf{X}}_s}) = \mathcal{N}(\mathbf{f}^{\tilde{\mathbf{X}}_s} | \mathbf{0}, \mathbf{k}_p^{\tilde{\mathbf{X}}_s, \tilde{\mathbf{X}}_s}), \tag{9}$$

with the kernel matrices $\mathbf{k}^{\tilde{\mathbf{X}}_s, \tilde{\mathbf{X}}_s}, \mathbf{k}_p^{\tilde{\mathbf{X}}_s, \tilde{\mathbf{X}}_s} \in \mathbb{R}^{|\tilde{\mathbf{X}}_s|C \times |\tilde{\mathbf{X}}_s|C}$ as the joints of pair-wise outcomes. Thus, the marginal KL divergence and its gradients can be estimated exactly without resorting to complicated approximations (Sun et al., 2019; Rudner et al., 2021).

Moreover, as discussed in Section 4.2, $k_p(\boldsymbol{x}, \boldsymbol{x}') = \hat{k}_p(\boldsymbol{x}, \boldsymbol{x}') \mathbf{I}_C$, so we can write $\mathbf{k}_p^{\tilde{\mathbf{X}}_s, \tilde{\mathbf{X}}_s}$ by Kronecker product: $\mathbf{k}_p^{\tilde{\mathbf{X}}_s, \tilde{\mathbf{X}}_s} = \hat{\mathbf{k}}_p^{\tilde{\mathbf{X}}_s, \tilde{\mathbf{X}}_s} \otimes \mathbf{I}_C$, where $\hat{\mathbf{k}}_p^{\tilde{\mathbf{X}}_s, \tilde{\mathbf{X}}_s} \in \mathbb{R}^{|\tilde{\mathbf{X}}_s| \times |\tilde{\mathbf{X}}_s|}$ corresponds to the evaluation of kernel $\hat{k}_p$. Hence in the computation of the KL, we can exploit the property of Kronecker product to inverse $\mathbf{k}_p^{\tilde{\mathbf{X}}_s, \tilde{\mathbf{X}}_s}$ in $\mathcal{O}(|\tilde{\mathbf{X}}_s|^3)$ complexity. Besides, as $\mathbf{k}^{\tilde{\mathbf{X}}_s, \tilde{\mathbf{X}}_s}$ is low-rank, we can leverage the matrix determinant lemma (Harville, 1998) to compute the determinant of $\mathbf{k}^{\tilde{\mathbf{X}}_s, \tilde{\mathbf{X}}_s}$ in $\mathcal{O}(|\tilde{\mathbf{X}}_s|CM^2)$ time given that usually $M \ll |\tilde{\mathbf{X}}_s|C$ (e.g., $10 \ll 256C$).

**A qualified prior kernel** When facing high-dimensional data, we usually have no prior knowledge of them but only know that NNs are a good model family. In this case, NN-GPs (Neal, 1996) can be good priors – they correspond to the commonly used Gaussian priors on weights while carrying valuable inductive bias of specific NN architectures. However, the analytical estimation of NN-GPs is prohibitive when the NN architecture is deep, so we advocate using the MC estimates of NN-GPs (MC NN-GPs) (Novak et al., 2018) as an alternative. They are more accessible and have been proven effective by Wang et al. (2019).

Concretely, given an NN composed of a feature extractor $h(\cdot, \hat{\boldsymbol{w}}) : \mathcal{X} \to \mathbb{R}^{\hat{C}}$ and a linear readout layer, we denote the Gaussian prior on the weights of $h$ by $\mathcal{N}(\mathbf{0}, \text{diag}(\hat{\boldsymbol{\sigma}}^2))$ and the prior variance of the readout layer by $\sigma_w^2$ (for weights) and $\sigma_b^2$ (for bias). Then the corresponding MC NN-GP kernel is

$$k_p(\boldsymbol{x}, \boldsymbol{x}') := \hat{k}_p(\boldsymbol{x}, \boldsymbol{x}') \mathbf{I}_C \text{ with } \hat{k}_p(\boldsymbol{x}, \boldsymbol{x}') := \sigma_b^2 + \frac{\sigma_w^2}{\hat{S}\hat{C}} \sum_{s=1}^{\hat{S}} h(\boldsymbol{x}, \hat{\boldsymbol{w}}_s)^\top h(\boldsymbol{x}', \hat{\boldsymbol{w}}_s). \tag{10}$$

$\hat{\boldsymbol{w}}_s$ are i.i.d. samples from $\mathcal{N}(\mathbf{0}, \text{diag}(\hat{\boldsymbol{\sigma}}^2))$. As shown, $\hat{S}$ forward passes of $h$ are needed to evaluate the MC NN-GP prior. $h$'s weights are randomly generated, and its architecture can be freely chosen, not necessarily identical to that of $g$. In practice, DE-GP benefits from the learning in the parametric family specified by NN ensemble, and hence can frequently outperform the analytical NN-GP posteriors, which are even intractable for deep NN architectures, on some metrics.

### 4.4 Discussion

**Calibrated uncertainty** As the prior $p(f)$ has nondegenerate covariance, the KL in Eq. (8), which aligns $q(f)$ with $p(f)$, will prevent the functional inconsistency among the ensemble members from collapsing. Meanwhile, the expected log-likelihood in Eq. (8) enforces each ensemble member to yield the same, correct outcomes for the training data. Thereby, the model is able to yield relatively high functional inconsistency (i.e., uncertainty) for regions far away from the training data (see Fig. 1). By contrast, DE's uncertainty on these regions is shaped by uncontrollable randomness.

**Efficiency** Compared to the overhead introduced by DNNs, the effort for estimating the KL in Eq. (8) is negligible. The added cost of DE-GP primarily arises from the introduction of the extra measurement points and the evaluation of prior kernels. In practice, we use a small batch size for the extra measurement points. When using the MC NN-GP prior, we set it using cheap architectures and perform MC estimation in parallel. Thereby, DE-GP is only marginally slower than DE.

**Weight sharing** DE-GP does not care about how $g_i$ are parameterized, so we can perform weight sharing among $g_i$, for example, using a shared feature extractor and $M$ independent MLP classifiers to construct $M$ ensemble members (Deng et al., 2021). With shared weights, DE-GP is still likely to be reliable because our learning principle induces diversity in function space. Experiments in Section 5.4 validate this.

**Limitations** DE-GP loses parallelisability, but this is not a specific issue of DE-GP, e.g., the variants of DE based on POVI also entail concurrent updates of the ensemble members (D'Angelo & Fortuin, 2021).

## 5 Experiments

We perform extensive evaluation to demonstrate that DE-GP yields better uncertainty estimates than the baselines, while preserving non-degraded predictive performance. Given that the original Deep Ensemble itself is very performant in terms of accuracy and uncertainty quantification (Ovadia et al., 2019), we mainly focus on comparing to it and its popular variants, including DE, rDE, NN-GP, RMS, etc. Unless specified otherwise, we adopt MC NN-GPs with $\hat{S} = 10$ as the priors, where the weight and bias variance at each layer are $2/\texttt{fan\_in}$ and 0.01 ($\texttt{fan\_in}$ is the input dimension according to (He et al., 2015)). We set the sampling distribution for extra measurement points $\nu$ as the uniform distribution over the data region and leave the adoption of more complicated ones (Hafner et al., 2020) to future work. The number of MC samples for estimating the expected log-likelihood (i.e., $U$ in Algorithm 1) is 256. We set the regularization constant $\lambda$ as 0.05 times of the average eigenvalue of the central covariance matrices. More details are in Appendix.

### 5.1 Illustrative 1-D Regression

We build a regression problem with 8 data from $y = \sin 2x + \epsilon, \epsilon \sim \mathcal{N}(0, 0.2)$ as shown in Fig. 1. For NN-GP, we perform analytical GP regression without training DNNs. For DE-GP, DE, and rDE, we train 50 MLPs. By default, we set $\alpha = 1$.

Fig. 1 presents the comparison on prediction with the training efficiency comparison deferred to Appendix. As shown, DE-GP delivers calibrated uncertainty estimates across settings with only marginally added overheads upon DE. DE-GP is consistent with the gold standard NN-GP. Yet, DE and rDE suffer from degeneracy issue as the dimension of weights grows. We also clarify that NN-GP would face non-trivial scalability issues when handling deep architectures (Novak et al., 2018), while DE-GP bypasses them.

### 5.2 UCI Regression

We then assess DE-GP on 5 UCI real-valued regression problems. The used architecture is a MLP with 2 hidden layers of 256 units and ReLU activation. 10 networks are trained for DE, DE-GP and other variants. For DE-GP, we tune $\alpha$ according to validation sets.

We perform cross validation with 5 splits. Fig. 2 shows the results. DE-GP surpasses or approaches the baselines across scenarios in aspects of both test negative log-likelihood (NLL) and test root mean square error (RMSE). DE-GP even beats NN-GP, which is probably attributed to that the variational family specified

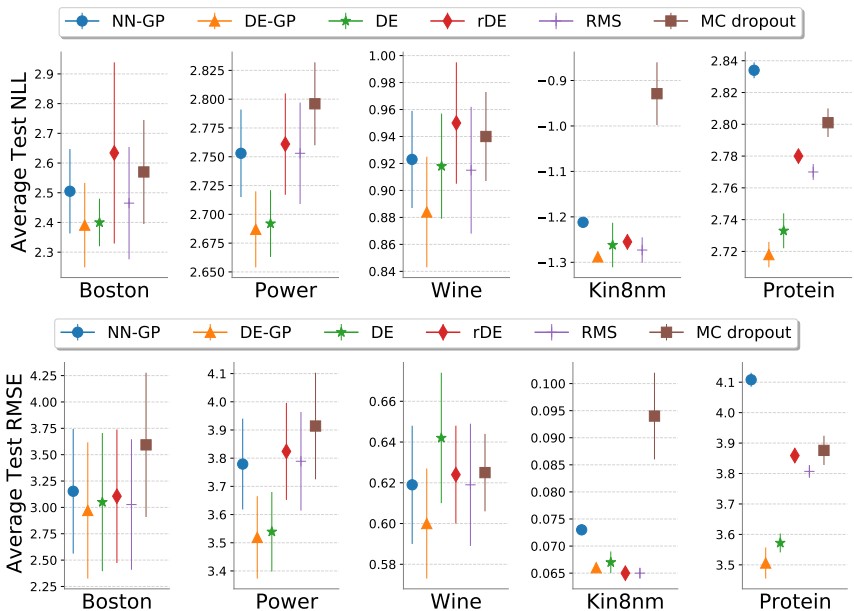

Figure 2: Average test NLL and RMSE on UCI regression problems. The lower the better.

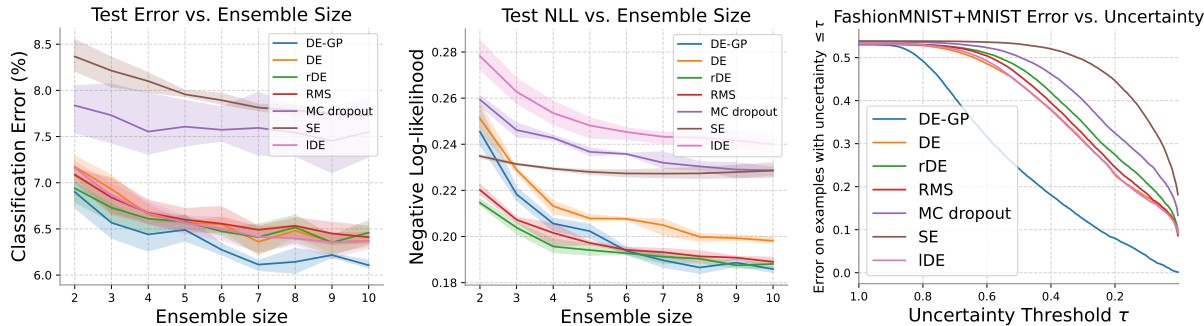

Figure 3: (Left): Test error varies w.r.t. ensemble size on Fashion-MNIST. (Middle): Test NLL varies w.r.t. ensemble size. (Right): Test error versus uncertainty plots for methods trained on Fashion-MNIST and tested on both Fashion-MNIST and MNIST. Ensemble size is 10.

by DE enjoys the beneficial inductive bias of practically sized SGD-trained DNNs, and DE-GP can flexibly trade off between the likelihood and the prior by tuning $\alpha$.

### 5.3 Classification on Fashion-MNIST

We use a widened LeNet5 architecture with batch normalizations (BNs) (Ioffe & Szegedy, 2015) for the Fashion-MNIST dataset (Xiao et al., 2017). Considering the inefficiency of NN-GP, we mainly compare DE-GP to DE, rDE, RMS, and MC dropout. We also include two other baselines: snapshot ensemble (SE) (Huang et al., 2016), which collects ensemble members from the SGD trajectory, and a variant of DE that performs the aggregation in the logit space (lDE). We set $\alpha$ as well as the regularization coefficients for rDE and RMS all as 0.1 according to validation accuracy. We augment the data log-likelihood (i.e., the first term in Eq. (8)) with a trainable temperature to tackle oversmoothing and avoid underconfidence.

The in-distribution performance is averaged over 8 runs. Fig. 3-(Left) & (Middle) display how ensemble size impacts the test results. We see the test error of DE-GP is lower than the baselines, and its test NLL decreases rapidly as the ensemble size increases.

Table 1: Test and NLL accuracy comparison on CIFAR-10. Results are summarized over 8 trials.

| | ResNet-20 | | ResNet-56 | |
| | Accuracy | NLL | Accuracy | NLL |
|---|---|---|---|---|
| DE-GP ($\beta = 0.1$) | **94.67**±0.04% | **0.164**±0.002 | **95.55**±0.04% | **0.148**±0.003 |
| DE-GP ($\beta = 0$) | 93.71±0.06% | 0.196±0.001 | 94.24±0.07% | 0.195±0.007 |
| DE | 93.43±0.08% | 0.214±0.001 | 94.04±0.07% | 0.197±0.002 |
| rDE | **94.58**±0.05% | **0.166**±0.001 | **95.56**±0.06% | **0.146**±0.001 |
| RMS | 93.63±0.07% | 0.201±0.001 | 94.45±0.03% | 0.179±0.001 |
| MC dropout | 92.38±0.02% | 0.316±0.007 | 93.63±0.16% | 0.324±0.015 |
| SE | 92.44±0.26% | 0.310±0.007 | 93.92±0.11% | 0.285±0.010 |
| lDE | 94.63±0.09% | 0.211±0.001 | 95.55±0.06% | 0.208±0.002 |

Besides, to compare the quality of uncertainty estimates, we use the trained models to make prediction and quantify epistemic uncertainty for both the in-distribution test set and the out-of-distribution (OOD) MNIST test set. All predictions on OOD data are regarded as wrong. The epistemic uncertainty is estimated by the mutual information between the prediction and the variable function:

$$\mathcal{I}(f, y | \boldsymbol{x}, \mathcal{D}) \approx H\left(\frac{1}{S} \sum_{s=1}^{S} p(y | f_s(\boldsymbol{x}))\right) - \frac{1}{S} \sum_{s=1}^{S} H\left(p(y | f_s(\boldsymbol{x}))\right), \tag{11}$$

where $H$ indicates Shannon entropy, with $f_s = g(\cdot, \boldsymbol{w}_s)$ for DE, rDE, and RMS, and $f_s \sim q(f; \boldsymbol{w}_1, ..., \boldsymbol{w}_M)$ for DE-GP. This is a naive extension of the weight uncertainty-based mutual epistemic uncertainty. We normalize the uncertainty estimates into $[0, 1]$. For each threshold $\tau \in [0, 1]$, we plot the average test error for data with $\leq \tau$ uncertainty in Fig. 3-(Right). We see under various uncertainty thresholds, DE-GP makes fewer mistakes than baselines, implying DE-GP can assign relatively higher uncertainty for the OOD data. We defer the performance of uncertainty-based OOD detection to the appendix.

### 5.4 Classification on CIFAR-10

Next, we apply DE-GP to the real-world image classification task CIFAR-10 (Krizhevsky et al., 2009). We consider the popular ResNet (He et al., 2016) architectures including ResNet-20 and ResNet-56. The ensemble size is fixed as 10. We split the data as the training set, validation set, and test set of size 45000, 5000, and 10000, respectively. We set $\alpha = 0.1$ according to an ablation study in Appendix. We use a lite ResNet-20 architecture without BNs and residual connections to set up the MC NN-GP prior kernel for both the ResNet-20 and ResNet-56-based variational posteriors.

Ideally, the KL divergence in Eq. (8) is enough to help DE-GP to resist over-fitting (in function space). Its effectiveness is evidenced by the above results, but we have empirically observed that it may lose efficacy in the CIFAR-10 + ResNets case. Specifically, we inspect how the $L_2$ norm of weights of the NN ensemble varies w.r.t. training step. We include three more approaches into the comparison: the DE-GP equipped with an extra $L_2$ regularization on weights with coefficient $\beta = 0.1$,[3] DE, and rDE. The results are displayed in Fig. 4 where DE-GP ($\beta = 0$) refers to the vanilla DE-GP. The generalization performance of these four approaches can be found in Table 1. An immediate conclusion is that the KL divergence in DE-GP's learning objective cannot cause proper regularization effects on weights, so the learned DE-GP suffers from high complexity and hence poor performance. This is probably caused by the high non-linearity of ResNets. We then advocate

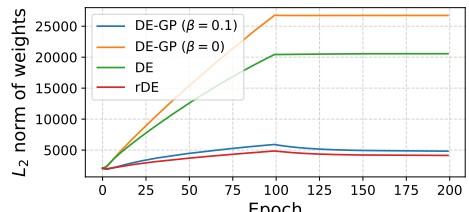

Figure 4: The $L_2$ norm of weights varies w.r.t. training step. The models are trained on CIFAR-10 with ResNet-20 architecture. DE-GP ($\beta = 0$) finds solutions with high complexity and poor test accuracy (see Table 1), yet DE-GP ($\beta = 0.1$) settles this.

---

[3]We set $\beta = 0.1$ without explicit tuning – just make it equivalent to the regularization coefficient on weight in rDE.

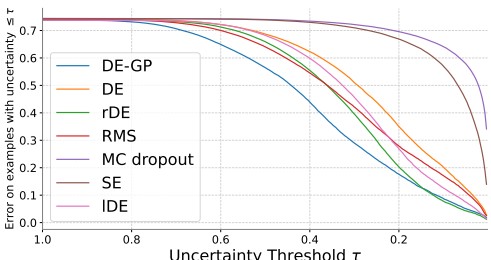 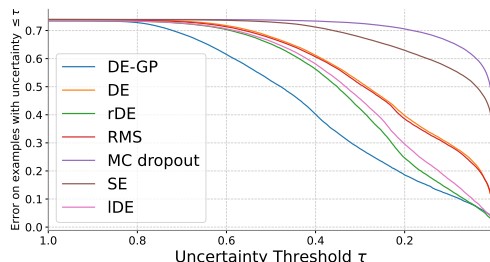

Figure 5: Test error versus uncertainty plots for methods trained on CIFAR-10 and tested on both CIFAR-10 and SVHN with ResNet-20 (Left) or ResNet-56 (Right) architecture.

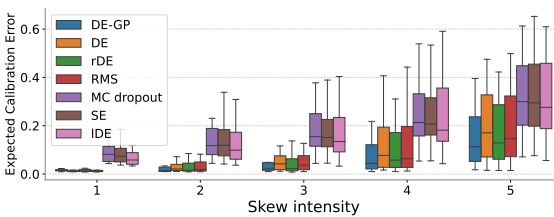 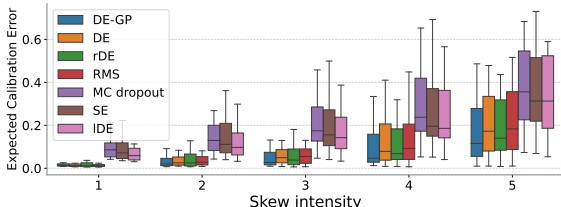

Figure 6: Expected Calibration Error on CIFAR-10 corruptions for models trained with ResNet-20 (Left) or ResNet-56 (Right). We summarize the results across 19 types of skew in each box.

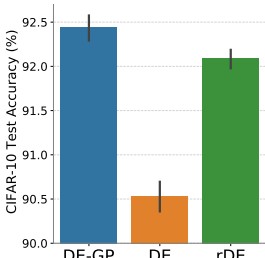 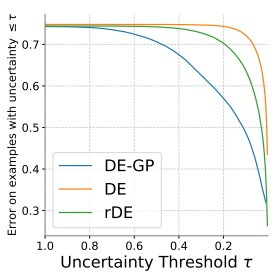 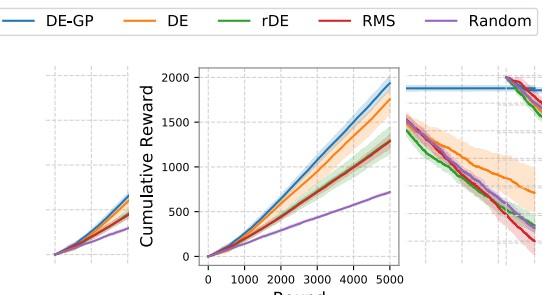

Figure 7: In-distribution test accuracy (Left) and error versus uncertainty plots on CIFAR-10+SVHN (Right) under weight sharing. (ResNet-20)

Figure 8: Cumulative reward varies w.r.t. round on Covertype. Random corresponds to the Uniform algorithm. Summarized over 5 trials.

explicitly penalizing the $L_2$ norm of the weights $\sum_i ||\boldsymbol{w}_i||_2^2$ to guarantee the generalization performance when applying DE-GP to handling deep architectures. This extra weight-space regularization may introduce bias to the posterior inference corresponding to the imposed prior. But, if we think of it as a kind of extra prior knowledge, we can then justify it within the posterior regularization scheme (Ganchev et al., 2010) (see Appendix).

In the following, we abuse DE-GP to represent DE-GP ($\beta = 0.1$) if there is no misleading. Fig. 5 shows the error versus uncertainty plots on the combination of CIFAR-10 and SVHN test sets. The results are similar to those for Fashion-MNIST.

We further test the trained methods on CIFAR-10 corruptions (Hendrycks & Dietterich, 2018), a challenging OOD generalization/robustness benchmark for deep models. As shown in Fig. 6 and Appendix, DE-GP reveals smaller Expected Calibration Error (ECE) (Guo et al., 2017) and lower NLL at various levels of skew, reflecting its ability to make conservative predictions under corruptions.

More results for the deeper ResNet-110 architecture and the more challenging CIFAR-100 benchmark are provided in Appendix.

Table 2: Ablation study on $\alpha$ for DE-GP ($\beta = 0.1$) (using ResNet-20 on CIFAR-10).

| $\alpha$ | 0.1 | 0.05 | 0.01 | 0.005 |
|---|---|---|---|---|
| Accuracy | 94.67±0.09% | 94.66±0.07% | 94.67±0.04% | 94.83±0.10% |

Table 3: Ablation study on the architecture of the prior MC NN-GP kernel.

| DE-GP architecture \ Prior kernel architecture | ResNet-20 | ResNet-56 | ResNet-110 |
|---|---|---|---|
| ResNet-56 (10 ensemble member) | 95.50% | 95.28% | - |
| ResNet-110 (5 ensemble member) | 95.54% | - | 94.87% |

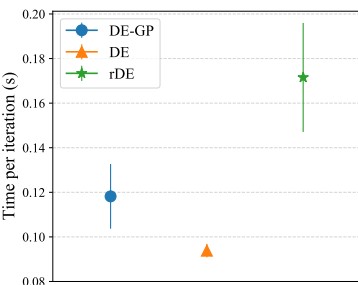

Figure 9: Training speed comparison on illustrative regression.

**Weight sharing** We build a ResNet-20 with 10 classification heads and a shared feature extraction module to evaluate the methods under weight sharing. We set a larger value for $\alpha$ for DE-GP to induce higher magnitudes of functional inconsistency. The test accuracy (over 8 trials) and error versus uncertainty plots on CIFAR-10 are illustrated in Fig. 7. We exclude RMS from the comparison as it assumes i.i.d. ensemble members which may be incompatible with weight sharing. DE-GP benefits from a functional inconsistency-promoting term (see Section 4.4), hence performs better than DE and rDE, which purely hinge on the randomness in weight.

## 5.5 Contextual Bandit

Finally, we apply DE-GP to the contextual bandit, an important decision-making task where the uncertainty helps to guide exploration. Following (Osband et al., 2016), we use DE-GP to achieve efficient exploration inspired by Thompson sampling. We reuse most of the settings for UCI regression (see Appendix). We leverage the GenRL library to build a contextual bandit problem, Covertype (Riquelme et al., 2018). The cumulative reward is depicted in Fig. 8. As desired, DE-GP offers better uncertainty estimates and hence beats the baselines by clear margins. The potential of DE-GP in more reinforcement learning and Bayesian optimization scenarios deserves future investigation.

## 5.6 More Analyses

**Ablation study on $\alpha$** We have conducted an ablation study on $\alpha$ (using ResNet-20 on CIFAR-10). The results are presented in Table 2. We can see that DE-GP is not sensitive to the value of $\alpha$. We in practice set $\alpha = 0.1$ in the CIFAR experiments. We did not use a smaller $\alpha$ as it may result in colder posteriors and in turn worse uncertainty estimates.

**Ablation study on the architecture of prior kernel** We perform an ablation study on the architecture for defining the prior MC NN-GP kernel, with the results listed in Table 3. Surprisingly, using the cheap ResNet-20 architecture results in DE-GP with better test accuracy. We deduce this is because a deeper prior architecture induces a more complex, black-box correlation for the function, which may lead to over-regularization.

**Comparison on training cost** We present the training speed comparison in Fig. 9. As shown, DE-GP consumes only marginally added training cost than the standard DE.

# 6 Conclusion

In this work, we address the unreliability issue of the uncertainty of Deep Ensemble by defining a Gaussian process with Deep Ensemble and training the model under the principle of functional variational inference. Doing so, we have successfully calibrated the uncertainty of the ensemble of NNs. We offer recipes to make the training feasible, and further identify some empirical characteristics of DE-GP. Our method can be implemented easily and efficiently. Extensive experiments validate the effectiveness of our method. We hope this work may shed light on the development of better Bayesian deep learning approaches.

**Broader Impact Statement**

This work proposes a Bayesian refinement of the Deep Ensemble. Its potential positive impacts on society are evident: its ability to enable better uncertainty estimation while maintaining predictive performance is crucial in the industry, e.g., automatic driving, disease analysis, and financial applications. In this scenario, the uncertainty estimates could be used to reject uncertain predictions and raise the requirement of inviting humans into the decision process. As a fundamental research in machine learning, the negative consequences are not obvious. Though in theory any technique can be misused, it is not likely to happen currently.

**Acknowledgments**

This work was supported by NSF of China (No. 62306176), Key R&D Program of Shandong Province, China (2023CXGC010112), Natural Science Foundation of Shanghai (No. 23ZR1428700), and CCF-Baichuan-Ebtech Foundation Model Fund.

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

# A APPENDIX

## A.1 An Explanation for the Weight-space Regularization

Posterior regularization (Ganchev et al., 2010; Zhu et al., 2014) provides a workaround for Bayesian approaches to impose extra prior knowledge. We can apply posterior regularization to functional variational inference by solving:

$$\max_{q(f)} \mathcal{L} = \mathbb{E}_{q(f)}[\log p(\mathcal{D}|f)] - D_{\mathrm{KL}}[q(f)\|p(f)] \text{ s.t. } q(f) \in Q. \tag{12}$$

$Q = \{q(f)|\mathbb{E}_{q(f)}\Omega(f) \leq 0\}$ is a valid set defined in terms of a functional $\Omega$ which delivers some statistic of interest of a function.[4] For tractable optimization, we can slack the constraint as a penalty:

$$\max_{q(f)} \mathcal{L}' = \mathcal{L} - \beta \max\{\mathbb{E}_{q(f)}\Omega(f), 0\}, \tag{13}$$

where $\beta$ is a trade-off coefficient.

We next show that the extra weight-space regularization can be derived by imposing the extra prior that functions drawn from DE-GP should generalize well to the learning of DE-GP given the above paradigm.

**Binary classification** In the binary classification scenario, $y \in \{-1, 1\}$ and $f, g_i : \mathcal{X} \to \mathbb{R}$. We use 0-1 loss $\ell(f(\boldsymbol{x}), y) = \mathbf{1}_{y \neq \mathrm{sign}(f(\boldsymbol{x}))}$ to measure the classification error on one datum. We assume an underlying distribution $\mu = \mu(\boldsymbol{x}, y)$ supported on $\mathcal{X} \times \{-1, 1\}$ for generating the training data $\mathcal{D}$, based on which we can define the true risk of a function (hypothesis) $f$: $R(f) := \mathbb{E}_{(\boldsymbol{x}, y) \sim \mu} \ell(f(\boldsymbol{x}), y)$. We set $\mathbb{E}_{q(f)}\Omega(f) := \mathbb{E}_{q(f)}R(f)$ in the seek of a posterior over functions that can generalize well.

By definition, a hypothesis sample $f \sim q(f) = \mathcal{GP}(m(\boldsymbol{x}), k(\boldsymbol{x}, \boldsymbol{x}'))$ can be decomposed as $f(\boldsymbol{x}) = \frac{1}{M}\sum_{i=1}^{M} g_i(\boldsymbol{x}) + \zeta(\boldsymbol{x})$ with $\zeta(\boldsymbol{x}) \sim \mathcal{GP}(0, k(\boldsymbol{x}, \boldsymbol{x}'))$. If $\mathrm{sign}(f(\boldsymbol{x})) \neq y$, it is impossible that $\mathrm{sign}(g_1(\boldsymbol{x})) = y$, ..., $\mathrm{sign}(g_M(\boldsymbol{x})) = y$, and $\mathrm{sign}(\zeta(\boldsymbol{x})) = y$ all hold. In other words,

$$\ell(f(\boldsymbol{x}), y) \leq \sum_{i=1}^{M}[\ell(g_i(\boldsymbol{x}), y)] + \ell(\zeta(\boldsymbol{x}), y). \tag{14}$$

We can further re-parameterize $\zeta(\boldsymbol{x})$ as $\zeta(\boldsymbol{x}) = \frac{1}{\sqrt{M}}\sum_{i=1}^{M} \epsilon_i(g_i(\boldsymbol{x}) - m(\boldsymbol{x})) + \sqrt{\lambda}\epsilon_0, \epsilon_i \sim \mathcal{N}(0, 1), i = 0, ..., M$, which is essentially a real-valued random function symmetric around 0. Thus, for any $(\boldsymbol{x}, y) \sim \mu$, we have $\mathbb{E}_{q(f)}\ell(\zeta(\boldsymbol{x}), y) = \mathbb{E}_{\epsilon_0, ..., \epsilon_M}\ell(\zeta(\boldsymbol{x}), y) = 1/2$. As a result,

$$\mathbb{E}_{q(f)}R(f) \leq \mathbb{E}_{q(f)}\mathbb{E}_{(\boldsymbol{x}, y) \sim \mu} \sum_{i=1}^{M}[\ell(g_i(\boldsymbol{x}), y)] + \mathbb{E}_{(\boldsymbol{x}, y) \sim \mu}[1/2] = \sum_{i=1}^{M}[R(g_i)] + 1/2. \tag{15}$$

Namely, the expected generalization error of the approximately posteriori functions can be bounded from above by those of the DNN basis functions. Recalling the theoretical and empirical results showing that DNNs' generalization error $R(g_i)$ can be decreased by controlling model capacity in terms of norm-based regularization $\min_{\boldsymbol{w}_i} \|\boldsymbol{w}_i\|_2^2$ (Neyshabur et al., 2015; 2017; Bartlett et al., 2017; Jiang et al., 2019), we obtain an explanation for the extra weight-space regularization.

**Multi-class Classification** In the multi-class classification scenario where $y \in \{1, 2, ..., C\}$ and $f, g_i : \mathcal{X} \to \mathbb{R}^C$, we use the loss $\ell(f(\boldsymbol{x}), y) = \mathbf{1}_{f(\boldsymbol{x})[y] < \max_{y' \neq y} f(\boldsymbol{x})[y']}$ to measure prediction error where $f(\boldsymbol{x})[j]$ denotes $j$-th coordinate of $f(\boldsymbol{x})$. The distinct difference between this scenario and the binary classification scenario is that in this setting, $\zeta(\boldsymbol{x})$ is a vector-valued function:

$$\zeta(\boldsymbol{x}) = \frac{1}{\sqrt{M}}\sum_{i=1}^{M} \epsilon_i(g_i(\boldsymbol{x}) - m(\boldsymbol{x})) + \sqrt{\lambda}\boldsymbol{\epsilon}_0, \tag{16}$$

where $\boldsymbol{\epsilon}_0 \sim \mathcal{N}(\mathbf{0}, \mathbf{I}_C)$ and $\epsilon_i \sim \mathcal{N}(0, 1), i = 1, ..., M$. We then make a mild assumption to simplify the analysis.

---

[4]Here we assume one-dimensional outputs for $\Omega$ for notation compactness.

**Assumption 1** *For any $(\boldsymbol{x}, y) \in \mu$, the elements on the diagonal of $k(\boldsymbol{x}, \boldsymbol{x})$ have the same value.*

This assumption implies that for any $j, j' \in \{1, ..., C\}$,

$$\frac{1}{M} \sum_{i=1}^{M} (g_i(\boldsymbol{x})[j] - m(\boldsymbol{x})[j])^2 + \lambda = \frac{1}{M} \sum_{i=1}^{M} (g_i(\boldsymbol{x})[j'] - m(\boldsymbol{x})[j'])^2 + \lambda. \tag{17}$$

I.e.,

$$\sum_{i=1}^{M} (g_i(\boldsymbol{x})[j] - m(\boldsymbol{x})[j])^2 = \sum_{i=1}^{M} (g_i(\boldsymbol{x})[j'] - m(\boldsymbol{x})[j'])^2. \tag{18}$$

Therefore, $\zeta(\boldsymbol{x})$ possesses the same variance across its output coordinates and becomes a random guess classifier. Based on this, we have $\mathbb{E}_{q(f)} \ell(\zeta(\boldsymbol{x}), y) = \mathbb{E}_{\epsilon_0,...,\epsilon_M} \ell(\zeta(\boldsymbol{x}), y) = (C-1)/C$. We can then derive a similar conclusion to that in the binary classification.

**The validity of Assumption 1.** When the data dimension $|\mathcal{X}|$ is high and the number of training data $n$ is finitely large, with zero probability the sampled data $(\boldsymbol{x}, y) \sim \mu$ resides in the training set. Therefore, only the KL divergence term of the fELBO explicitly affects the predictive uncertainty at $\boldsymbol{x}$. Because the considered priors possess a diagonal structure, it is hence reasonable to make the above assumption.

## A.2 Detailed Experimental Settings

**Illustrative regression.** For the problem on $y = \sin 2x + \epsilon, \epsilon \sim \mathcal{N}(0, 0.1)$, we randomly sample 8 data points from $[-1.5, 1.5]$. We add $-1.2$ to the target value of the rightest data point to introduce strong data noise. For optimizing the ensemble members, we use a SGD optimizer with 0.9 momentum and 0.001 learning rate. The learning rate follows a cosine decay schedule. The optimization takes 1000 iterations. The extra measurement points are uniformly sampled from $[-2, 2]$. The regularization constant $\lambda$ is set as $1e-4$ times of the average eigenvalue of the central covariance matrices.

**UCI regression.** We pre-process the UCI data by standard normalization. We set the variance for data noise and the weight variance for the prior kernel following (Pearce et al., 2020). The batch size for stochastic training is 256. We use an Adam optimizer to optimize for 1000 epochs. The learning rate is initialized as 0.01 and decays by 0.99 every 5 epochs.

**Fashion-MNIST classification.** The used architecture is Conv(32, 3, 1)-BN-ReLU-MaxPool(2)-Conv(64, 3, 0)-BN-ReLU-MaxPool(2)-Linear(256)-ReLU-Linear(10), where Conv($x$, $y$, $z$) represents a 2D convolution with $x$ output channels, kernel size $y$, and padding $z$. The batch size for training data is 64. We do not use extra measurement points here. We use an SGD optimizer to optimize for 24 epochs. The learning rate is initialized as 0.1 and follows a cosine decay schedule. We use an Adam optimizer with $1e-3$ learning rate to optimize the temperature. We use 1000 MC samples to estimate the posterior predictive and the epistemic uncertainty, because the involved computation is only the cheap softmax transformation on the sampled function values.

**CIFAR-10 classification.** We perform data augmentation including random horizontal flip and random crop. The batch size for training data is 128. We do not use extra measurement points here. We use a SGD optimizer with 0.9 momentum to optimize for 200 epochs. The learning rate is initialized as 0.1 and decays by 0.1 at 100-th and 150-th epochs. We use an Adam optimizer with $1e-3$ learning rate to optimize the temperature. We use 1000 MC samples to estimate the posterior predictive and the epistemic uncertainty. Suggested by Ovadia et al. (2019); He et al. (2020), we train models on CIFAR-10, and test them on the combination of CIFAR-10 and SVHN test sets. This is a standard benchmark for evaluating the uncertainty of OOD data.

**Contextual bandit.** We use MLPs with 2 hidden layers of 256 units. The batch size for training is 512. We do not use extra measurement points here. We update the model (i.e., the agent) for 100 epochs with an Adam optimizer every 50 rounds. We set $\alpha = 1$. DE, rDE, and RMS all randomly choose an ensemble member at per iteration, but our method randomly draws a sample from the defined Gaussian process for

decision. This is actually emulating Thompson Sampling and advocated by Bootstrapped DQN (Osband et al., 2016). "Random" baseline corresponds to the Uniform algorithm.

### A.3  More Experimental Results

We provide more results in this subsection.

### A.3.1  Results of DE-GP without Using Extra Measurement Points

We depict the results of DE-GP without using extra measurement points on $y = \sin 2x + \epsilon, \epsilon \sim \mathcal{N}(0, 0.2)$ problem in Fig. 10. The results are promising.

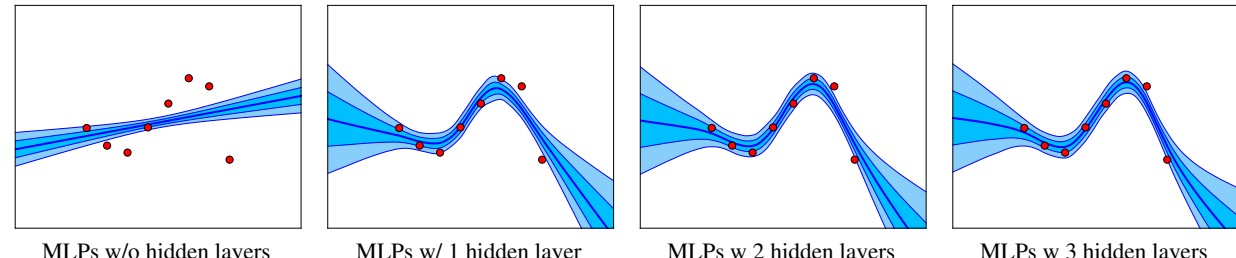

| MLPs w/o hidden layers | MLPs w/ 1 hidden layer | MLPs w 2 hidden layers | MLPs w 3 hidden layers |

Figure 10: Results of DE-GP without using extra measurement points. The settings are equivalent to those in the main text.

### A.3.2  Results Regarding OOD Detection

We present the results of uncertainty-based OOD detection in Table 4, which correspond to the Fashion-MNIST v.s. MNIST and CIFAR-10 v.s. SVHN scenarios in Fig. 3-(Right) and Fig. 5 respectively. As shown, our method exhibits the best AUROC.

Table 4: Uncertainty-based OOD detection results in terms of AUROC.

|       | Fashion-MNIST v.s. MNIST | CIFAR-10 v.s. SVHN (ResNet-56) |
|-------|:---:|:---:|
| DE-GP | **0.980** | **0.937** |
| DE    | 0.922 | 0.876 |
| rDE   | 0.907 | 0.931 |
| RMS   | 0.918 | 0.889 |
| lDE   | 0.921 | 0.934 |

### A.3.3  More Results on CIFAR-10 Classification

We plot the negative log-likelihood and test accuracy on CIFAR-10 corruptions for models trained with ResNet-20 and ResNet-56 in Fig. 11 and Fig. 12. As shown, DE-GP outperforms the baselines in the aspect of negative log-likelihood, but yields similar test accuracy to the baselines. Recapping the results in the main text, DE-GP indeed has improved OOD robustness, but may still face problems in OOD generalization.

We then conduct experiments with the deeper ResNet-110 architecture. Due to resource constraints, we use 5 ensemble members. The other settings are roughly the same as those for ResNet-56. The results are offered in Fig. 13, which validate the effectiveness of DE-GP for large networks.

### A.3.4  Results on CIFAR-100

We further perform experiments on the more challenging CIFAR-100 benchmark. We present the in-distribution test accuracy of DE-GP as well as the baselines in Table 5. We can see that DE-GP ($\beta = 0.1$) is still on par with rDE.

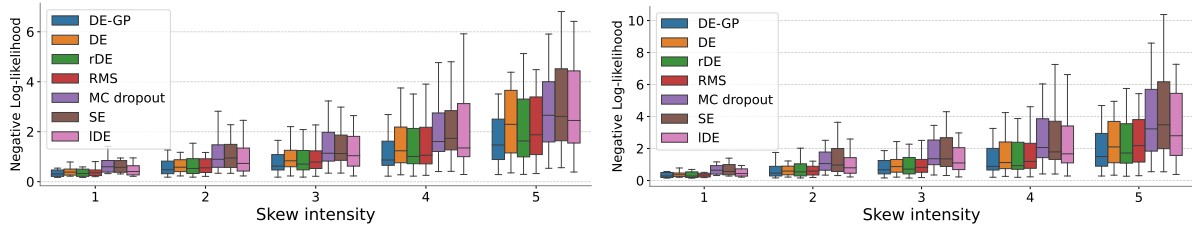

Figure 11: Negative log-likelihood on CIFAR-10 corruptions for models trained with ResNet-20 (Left) or ResNet-56 (Right) architecture. We summarize the results across 19 types of skew in each box.

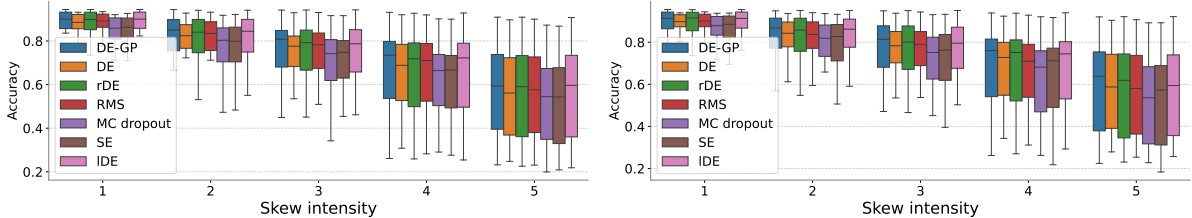

Figure 12: Test accuracy on CIFAR-10 corruptions for models trained with ResNet-20 (Left) or ResNet-56 (Right) architecture. We summarize the results across 19 types of skew in each box.

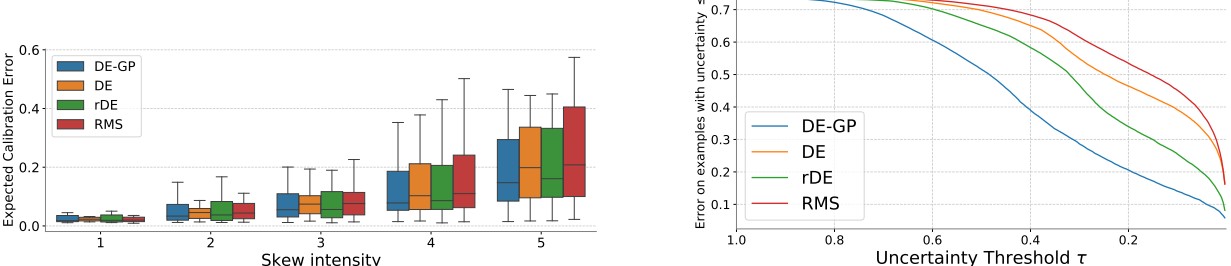

Figure 13: (Left): Expected Calibration Error on CIFAR-10 corruptions for models trained with ResNet-110 architecture. We summarize the results across 19 types of skew in each box. (Right): Test error versus uncertainty plots for methods trained on CIFAR-10 and tested on both CIFAR-10 and SVHN with ResNet-110 architecture. Ensemble size is fixed as 5 for these experiments.

Table 5: Test accuracy comparison on CIFAR-100.

| Architecture | DE-GP ($\beta = 0.1$) | DE | rDE | RMS |
|---|---|---|---|---|
| ResNet-20 | 76.59% | 74.14% | **76.81**% | 75.08% |
| ResNet-56 | **79.51**% | 76.46% | 79.21% | 76.77% |

We depict the error versus uncertainty plots on the combination of CIFAR-100 and SVHN test sets in Fig. 14. It is shown that the uncertainty estimates yielded by DE-GP for OOD data are more calibrated than the baselines.

We further test the trained methods on CIFAR-100 corruptions (Hendrycks & Dietterich, 2018), and present the comparisons in aspects of test accuracy and NLL in Fig. 15. It is evident that DE-GP reveals lower NLL than the baselines at various levels of skew.

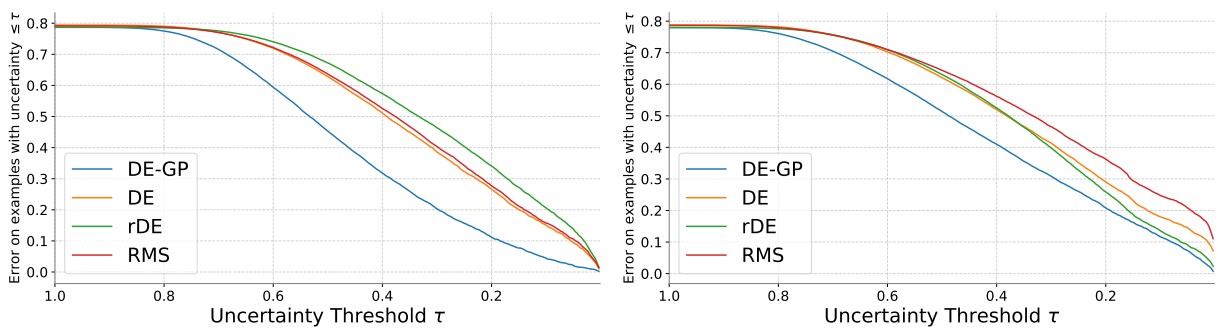

Figure 14: Test error versus uncertainty plots for methods trained on CIFAR-100 and tested on both CIFAR-100 and SVHN with ResNet-20 (Left) or ResNet-56 (Right) architecture.

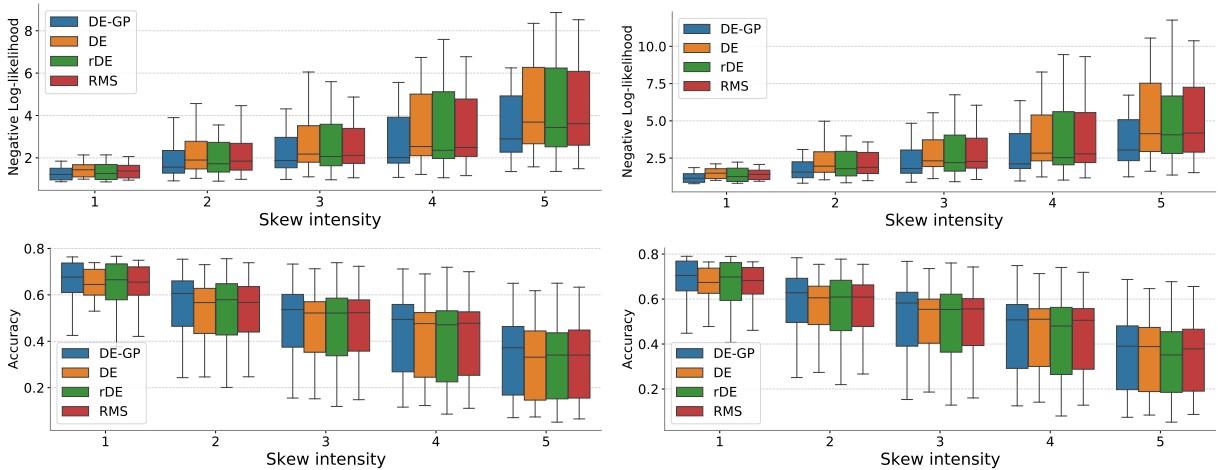

Figure 15: First row: test NLL on CIFAR-100 corruptions for models trained with ResNet-20 (Left) or ResNet-56 (Right) architecture. Second row: test accuracy on CIFAR-100 corruptions for models trained with ResNet-20 (Left) or ResNet-56 (Right) architecture. We summarize the results across 19 types of skew in each box.

### A.3.5 Results on TinyImageNet

We conduct experiments on TinyImageNet (mnmoustafa, 2017) and here are some core results (we ensemble 5 members with ResNet-18 architecture and perform training for 30 epochs with an SGD optimizer):

Table 6: Empirical comparison on TinyImageNet.

|        | Acc    | NLL  | ECE   |
|--------|--------|------|-------|
| rDE    | 49.6%  | 2.10 | 0.023 |
| DE-GP  | 49.3%  | 2.15 | 0.017 |

As shown, despite marginally worse accuracy and NLL, DE-GP is substantially more calibrated than rDE, hence enjoying better uncertainty estimates.

