# OpenReview forum: "Calibrating Deep Ensemble through Functional Variational Inference"
_TMLR — Accepted by TMLR_

### Review · Reviewer_Ym7q · 2024-06-05

**Summary Of Contributions:**

The authors propose to take functional inconsistency into account in the training stage of deep ensemble (DE) to prevent functional mode collapse by regularize the empirical GP defined by DE with a prior GP. The proposed method is evaluated over standard regression & classification, and contextual bandit benchmarks.

**Audience:**

Yes

**Claims And Evidence:**

Yes

**Requested Changes:**

See weakness

**Strengths And Weaknesses:**

Stength:
1. The motivation of the method is clear (i.e., prevent function space mode collapse) and the method seems natural and easy to implement (compared with some other function space Bayesian methods).
2. The performance of the method looks ok when comapred with the baselines considered in the paper.
3. The paper is well structured and is easy to follow.

Weakness:
1. Please consider reporting NLL for classification on in-distribution data as well (i.e., without corruptions)
2. The performance of OOD detection tasks is typically reported for Bayesian UQ paper, but is missing in this submission.
3. The baselines are not very strong. E.g. D’Angelo et al. 2021 is mentioned in the related work as a concurrent work but no comparison with it is done through experiments.

Minor:
1. The first sentence in the second paragraph in Section 1 "Deep Ensemble (DE) (Lakshminarayanan et al., 2017) is an effective and practical BNN method..." - this is arguable (as mentioned in you paper as well in the related work section). I suggest the authors consider rewording it a bit.
2. The KL[q(f)||p(f)] seems to be intractble as the approx. posterior and prios GP are based on different kernels (it seems to be a common issue for some other fVI papers as well though).

---

> ### Author Response · Authors · 2024-06-07
> **Thanks for your review**
>
> Thank you for taking the time to review our paper. We appreciate your recognition of the clear motivation, ease of implementation, acceptable performance, and satisfactory writing of the paper. We have carefully considered the concerns you raised and would like to address them individually below.
>
> ### Q1: Consider reporting NLL for classification on in-distribution data as well (i.e., without corruptions)
> Sorry for the missing results. We have updated Table 1 of the submission to include the NLL for the classification of in-distribution data on CIFAR-10. We observe that the NLLs are consistent with accuracy values, so the conclusion made in our paper still holds.
>
> ### Q2: Regarding the performance of OOD detection
> Sorry for the missing results. We first clarify that the error versus uncertainty plots in Fig. 3-(Right) and Fig. 5 are prevalent for measuring the quality of uncertainty estimates (see [1-3]) and they can somehow indicate the performance of uncertainty-based OOD detection. To prove this, we have added the results of distinguishing Fashion-MNIST from MNIST and CIFAR-10 from SVHN in Table 4 of Appendix A.3.2. The results verify the superiority of our method.
>
> [1] Simple and Scalable Predictive Uncertainty Estimation using Deep Ensembles.
>
> [2] Can You Trust Your Model’s Uncertainty? Evaluating Predictive Uncertainty Under Dataset Shift.
>
> [3] Uncertainty Estimation Using a Single Deep Deterministic Neural Network
>
>
> ### Q3: The baselines are not very strong. E.g. D’Angelo et al. 2021 is mentioned in the related work as a concurrent work but no comparison with it is done through experiments.
> Thanks for the kind suggestion. We clarify that because the original deep ensemble itself is very performant, we mainly focus on comparing to it and its popular variants. We have a total of 6 baselines in our main results on CIFAR-10: DE, rDE, RMS, MC dropout, SE, and lDE. Besides, on regression tasks, we have compared our method with strong baselines like NN-GP.
>
> Regarding the repulsive DE method [D’Angelo et al. 2021], we cannot perform a fair comparison with it because it seems to be incompatible with the batch normalizations (BNs) in ResNet (see [here](https://github.com/ratschlab/repulsive_ensembles/blob/master/experiments/exp_cifar.py#L70)). We have tried to hack the code to make it work for BN-ResNet but failed. This also reflects the superior practicability of our method over this baseline.
>
> ### Q4: Minor issues regarding wording
> Thanks. Addressed.
>
> ### Q5: The KL[q(f)||p(f)] seems to be intractble as the approx. posterior and prios GP are based on different kernels (it seems to be a common issue for some other fVI papers as well though).
> Yes. Nevertheless, the posterior and prior should routinely be defined with different kernels, and the lower bound technique empirically works for the learning of our approach.

---

### Review · Reviewer_tucJ · 2024-06-11

**Summary Of Contributions:**

The paper proposes DE-GP, a new method to quantify uncertainty in deep learning. To address the over-confidence and functional inconsistency the Deep Ensemble method has, DE-GP view each function in ensemble as a basis function. These basis functions are descrbied by a Gaussian Process. To train the model, the authors use a functional variational inference given specific prior beliefs. DE-GP is compared with standard baselines on standard benchmarks and show it's effectiveness.

**Audience:**

Yes

**Broader Impact Concerns:**

The authors stated the broader impact.

**Claims And Evidence:**

Yes

**Requested Changes:**

The paper is well-written and has good motivation with a solid method. Extensive experiments are done, though they are on small/toy datasets. I wonder if DE-GP works in a scaled-up setting like in CelebA or ImageNet. Also, it would be more impactful if authors compare SD-GP with more recent baseline methods.

**Strengths And Weaknesses:**

Strengths

- Motivation for proposing DE-GP
- Teaser figure showing that DE-GP addresses the over-confidence issue that DE has, and follows the NN-GP uncertainty
- Extensive experiments on a small dataset shows the effectiveness of DE-GP
- well-written paper

Weakenesses

- Experiments on a real dataset: Experimental results show that DE-GP works well on small datasets like MNIST or CIFAR10. One merits of Deep Ensemble over other uncertainty methods would be the scalability. In that sense, I wonder how DE-GP would work on a real dataset like CelebA or ImageNet (or TinyImageNet).

- The baselines that are compared with SD-GP are quite out-dated. I was wondering if there are more recent works that are related with Deep Ensemble, and if SD-GP still outperforms those recent methods in terms of uncertainty quantification.

- [A] have shown that achievements of Deep Ensemble can be replicated by a larger single model. Within the same setting, would SD-GP outperform a larger single model?

[A] Deep Ensembles Work, But Are They Necessary? NeurIPS, 2022.

---

> ### Author Response · Authors · 2024-06-17
> **Thanks for your review**
>
> Thank you for dedicating your time to reviewing our paper. We are grateful for your acknowledgment of the paper's clear motivation, extensive experiments, and satisfactory writing. We have thoroughly considered the concerns you raised and would like to address them individually as follows.
>
>
> ### Q1: Experiments on a real dataset like CelebA or ImageNet (or TinyImageNet).
> Thanks for the suggestion. We conduct new experiments on the recommended TinyImageNet and report some initial results here:
> | | Acc | NLL | ECE |
> |:------|:-------:|:------:|:------:|
> | rDE | 49.6% | 2.10 | 0.023 |
> | DE-GP |  49.3% | 2.15  | 0.017|
> We ensemble 5 members with ResNet-18 architecture and perform training for 30 epochs (due to the limited rebuttal period) with an SGD optimizer.  As shown, despite marginally worse accuracy and NLL, DE-GP is substantially more calibrated than rDE, hence enjoying better uncertainty estimates.
>
>
> ### Q2: If DE-GP still outperforms those recent methods in terms of uncertainty quantification.
> Thanks for the advice. We would like to clarify that the original deep ensemble itself is very performant in terms of accuracy and uncertainty quantification, so we mainly focus on comparing to it and its popular variants. We have a total of 6 baselines in our main results on CIFAR-10: DE, rDE, RMS, MC dropout, SE, and lDE. Besides, on regression tasks, we have compared our method with strong baselines like NN-GP.
>
> The repulsive DE method [D’Angelo et al. 2021] is a more recent method for improving deep ensemble. However, we cannot perform a fair comparison with it because it seems to be incompatible with the batch normalizations (BNs) in ResNet (see [here](https://github.com/ratschlab/repulsive_ensembles/blob/master/experiments/exp_cifar.py#L70)). We have tried to hack the code to make it work for BN-ResNet but failed. This also reflects the superior practicability of our method over this baseline.
>
>
> ### Q3: [A] have shown that achievements of Deep Ensemble can be replicated by a larger single model. Within the same setting, would DE-GP outperform a larger single model?
>
> Thanks for pointing out such an interesting work, from which we have gained a lot of meaningful insights.
>
> To the question, we have to acknowledge that we are also unclear if DE-GP could outperform a larger single model in the same setting because there are too many variation factors in such a comparison. However, we clarify that our DE-GP approach can exhibit more meaningful uncertainty quantification OOD data (as shown in Figs. 3, 5, and 6) than vanilla deep ensemble because of the incorporation of approximate Bayesian inference. Namely, we don’t just rely on the ensemble diversity for uncertainty quantification OOD data. This echoes the argument in paper [A]: “we show that ensemble diversity, by any metric, does not meaningfully contribute to an ensemble’s uncertainty quantification on out-of-distribution (OOD) data”. Namely, our approach forms a proper fix to deep ensemble for uncertainty quantification.

---

### Review · Reviewer_LjxS · 2024-06-14

**Summary Of Contributions:**

The paper proposes a method to enhance the uncertainty quantification in Deep Ensemble (DE) models by incorporating functional inconsistency among ensemble members using functional variational inference (fVI). Traditional DE models suffer from unreliable uncertainty due to randomness in initialization and optimization. By characterizing functional inconsistency with empirical covariance and defining a Gaussian process, the proposed approach explicitly manages uncertainty and achieves better calibration. This method outperforms existing DE and BNN methods across various benchmarks, providing more reliable uncertainty estimates with minimal additional computational cost.

**Audience:**

Yes

**Broader Impact Concerns:**

The broader impact concerns include potential misuse of the improved uncertainty quantification in critical decision-making systems, such as autonomous driving or medical diagnosis, leading to over-reliance on automated systems. Additionally, the computational resources required for these methods could contribute to environmental concerns and increase the digital divide.

**Claims And Evidence:**

Yes

**Requested Changes:**

1. Motivation not clear, maybe try a better toy example.
2. Figure 1 should be well explained, no detailed illustration for figure 1.

**Strengths And Weaknesses:**

1. " (i) DE and rDE collapse to a single model in the linear case (i.e., without hidden layers), because the loss surface is convex w.r.t. the model parameters;"  This is a extreme case, I don't see any motivation from this observation.
2. " (ii) DE and rDE reveal minimal uncertainty in in-distribution regions, although there is severe data noise; " what does this mean, which part of the figure show this?
3. I can accept the hypothesis that the uncertainty brought by DE cannot well represent the bayesian inference, but why and I can not see from the toy example in figure 1 because it show an extreme case. a 2 hidden layer to fit the simple linear training data, it is obvious to overfit. Please give more detailed motivation the explanation.
4. I don't understand the relationship between your observation and epistemic or aleatoric uncertainty, please explain.
5. I don't understand how to directly add a small scaled identity matrix λIC in section 4.1.

---

> ### Author Response · Authors · 2024-06-17
> **Thanks for your review**
>
> Thank you for taking the time to review our paper. We are glad you found our method empirically effective. We have responded to the detailed questions below.
>
>
>
> ### Q1: Regarding results in Figure 1 and the motivation of the paper.
> We thank the reviewer for the detailed comments. However, we would like to clarify that we have offered a detailed illustration for Figure 1 at the bottom of Section 3 and the corresponding caption. By carefully parsing Figure 1, we can clearly identify the unreliability issues of deep ensemble and motivate the proposed approach. **This is acknowledged by the other two reviewers**. In particular, Reviewer tucJ also recognizes Figure 1 as one strength of our paper: “*teaser figure showing that DE-GP addresses the over-confidence issue that DE has, and follows the NN-GP uncertainty*”. So, we kindly ask the reviewer to re-evaluate the motivation of our approach.
>
> Before replying to the details points, we also clarify that “the linear case” only refers to the leftmost plot in each subfigure of Figure 1. **It corresponds to using a linear base model for the ensemble instead of training model on linear training data**. As mentioned in the caption of Figure 1, we sample training data from a sine function so the data is not linear.
>
> Regarding the linear case in Figure 1, we apologize for the less detailed expressions. What we want to express is that deep ensemble is likely to fail due to the lack of managing the functional inconsistency among the ensemble members. Though this is an extreme case, it is still reasonable to question if the issue exists in other more practical cases.
>
> Regarding "(ii) DE and rDE reveal minimal uncertainty in in-distribution regions, although there is severe data noise", please refer to the “with 2 hidden layers” case in the subfigures. As shown, the ensemble members’ predictions of DE (i.e., the black lines in Fig 1(a)-(b)) overlap significantly in the region where the in-distribution training data (i.e., red points) is located, although there is severe noise in the training data ($y = sin 2x + \epsilon,  \epsilon \sim \mathcal{N} (0, 0.2)$). So, we make such arguments.
>
> ### Q2: Relationship between observation and epistemic or aleatoric uncertainty.
> Thanks for raising such an interesting question. We point out that deep ensemble and Bayesian neural network approaches are usually used to quantify epistemic uncertainty, which results from the lack of complete knowledge. This is echoed by Figure 1: in the region where the in-distribution training data (i.e., red points) is located, the model has knowledge so the epistemic uncertainty is relatively low. In contrast, in the other regions (the left and right ends within each plot), the epistemic uncertainty increases. The epistemic uncertainty of DE-GP is obviously better than DE and rDE, aligning with the gold standard NN-GP.
>
> Aleatoric uncertainty, on the other hand, arises from ambiguity in data and is usually handled by heteroscedastic neural networks or other approaches. Our paper does not cover that aspect.
>
>
> ### Q3: How to directly add a small-scale identity matrix λIC in section 4.1.
> Sorry for the unclearness. We can defined a constant matrix-valued kernel $k’(x, x’) = \lambda I_C$, so the final kernel is just $k_{final}(x, x’) = k(x, x’) + k’(x, x’)$. It is very simple in the code implementation. We will make it clearer in the revision.

---

### Decision · Action_Editor_BJDJ · 2024-07-31

**Recommendation:** Accept with minor revision

**Comment:**

Reviewers mainly raised concerns regarding (1) experiments on relatively small datasets and outdated baselines, and (2) issues on discussing deep ensemble approach. The concerns are largely addressed by author feedback, specifically the authors have made a good argument that deep ensemble, despite being published in 2017, remains to be a strong baseline for uncertainty quantification in 2024.

Please incorporate your experimental results in the author feedback to the final camera ready. Also it would be useful to clarify more about making better discussions regarding the relationship between deep ensemble and (approximate) Bayesian inference. Lastly, there has been works related to FVI, e.g., https://arxiv.org/abs/2312.17199 that the authors should consider discussing and comparing.

**Audience:**

ML researchers interested in (Bayesian) uncertainty estimation in neural networks.

**Claims And Evidence:**

The paper proposes a function-space VI method to improve deep ensemble methods. Specifically, the idea is to use the members in the deep ensemble as a finite set of basis functions, and then construct an approximate Gaussian process posterior with its kernel defined using these basis functions. The VI objective is constructed using the KL divergence between stochastic processes.

Experiments compared the proposed method with both GP and deep ensemble related methods, on regression, classification and contextual bandit tasks. Overall the proposed approach performs better than the baselines.